# Pharmacists’ Knowledge, Attitudes, and Practices Toward CGRP Inhibitors in Migraine Management: A Cross-Sectional Study

**DOI:** 10.3390/healthcare13172231

**Published:** 2025-09-06

**Authors:** Anwar Seraj Alfahmi, Lana Abdullah Alqarni, Lura Abdulrahman Alkhatabi, Fahad S. Alshehri

**Affiliations:** 1College of Pharmacy, Umm Al-Qura University, Makkah 24382, Saudi Arabia; anwaralfhmi@gmail.com (A.S.A.); alqarnilanaq2@gmail.com (L.A.A.); loura.alkhatabi@gmail.com (L.A.A.); 2Department of Pharmacology and Toxicology, College of Pharmacy, Umm Al-Qura University, Makkah 24382, Saudi Arabia

**Keywords:** CGRP inhibitors, migraine, pharmacists, knowledge, attitudes, practices, Saudi Arabia, preventive therapy

## Abstract

**Background:** Calcitonin gene-related peptide (CGRP) inhibitors represent a novel class of medications for the prevention and treatment of migraines. Given their expanding role in migraine management, pharmacists are increasingly expected to be knowledgeable about their use. However, limited data exist regarding pharmacists’ awareness and engagement with CGRP inhibitors in Saudi Arabia. **Objective:** To assess the knowledge, attitudes, and practices (KAP) of pharmacists in Saudi Arabia toward CGRP inhibitors for migraine management and identify areas for improvement. **Methods:** A cross-sectional survey was conducted from October 2024 to January 2025 among licensed pharmacists in Saudi Arabia using an online questionnaire distributed via social media platforms, pharmacy networks, and in-person outreach. The survey consisted of 26 questions across four sections: demographics, knowledge, attitudes, and practices. Descriptive statistics and regression analyses were performed to evaluate the data using SPSS version 29. **Results:** A total of 419 pharmacists participated (response rate: 98.8%). Most practiced in community pharmacies (90.5%) and the Western region (48.2%). Overall, pharmacists demonstrated moderate knowledge (54.7%), attitudes (55.6%), and practices (49.9%) toward CGRP inhibitors in migraine management. While 54.7% were familiar with CGRP inhibitors, only 37.2% were aware of relevant clinical guidelines, and 32.5% recognized adverse effects such as hypertension. Regarding attitudes, 32.2% believed CGRP inhibitors were superior to conventional therapies, but over 50% expressed neutral views across key items. Practice patterns revealed limited engagement, with more than half reporting neutral responses toward patient education, interdisciplinary collaboration, and safety monitoring. **Conclusions:** The study highlights moderate awareness but suboptimal application of CGRP inhibitor knowledge among pharmacists. Notable gaps exist in guideline familiarity, patient education, and interdisciplinary collaboration. Targeted training and inclusion of CGRP inhibitors in pharmacy curricula and continuing education programs are warranted to support effective migraine management.

## 1. Introduction

Migraine is a chronic neurological disorder characterized by frequent attacks of moderate to severe headache associated with symptoms such as nausea, vomiting, photophobia, and phonophobia [1]. Migraine remains not fully understood; however, it is believed to be a result of the combination of neurological, genetic, and environmental predispositions [2,3,4]. The current understanding of migraine pathophysiology supports a theory considering vascular and neuronal factors [5]. Theory implies that the sensitization of sensory neurons in the trigeminal nucleus caudalis is responsible for the dilation and neurogenic inflammation of the meningeal blood vessels, which leads patients to experience acute attacks of throbbing, unilateral, and intense pain [2,5].

According to the World Health Organization (WHO), migraines are ranked as the second-most disabling disorder and third-most prevalent neurological disorder [6]. Moreover, women are approximately three times more likely to experience migraine attacks than men, a disparity largely attributed to hormonal fluctuations [7]. Migraines are a major health concern, affecting more than one billion individuals worldwide [8]. Migraines are one of the heaviest burdens on the affected individual, as they usually develop in the highest productive years (25–60 years) [9]. It is also considered to be disabling and devastating as the attack lasts between one day to one week and sometimes beyond that [9]. The WHO, through the Global Burden of Disease study, assesses the impact of migraines using indicators such as Disability-Adjusted Life Years (DALYs) and Years Lived with Disability (YLDs). Between 1990 and 2016, Saudi Arabia demonstrated one of the highest rising trends in age-standardized YLD rates for migraine and tension-type headache within the Eastern Mediterranean Region [10]. The prevalence of all types of headaches in Saudi Arabia is estimated to be 77.2%, with approximately 25% prevalence of migraine [11].

A recent study showed the proportion of migraines in Saudi Arabia is approximately 22% per 55,061 participants, which is considered both concerning and challenging to manage [11]. The pharmacological management of migraine includes both acute and preventive therapies [12]. Acute therapies, such as acetaminophen, non-steroidal anti-inflammatory drugs (NSAIDs), triptans, ergotamine, and some CGRP aim to rapidly relieve the pain [12]. Despite the advances in treatments, over-the-counter (OTC) analgesics are commonly used even with their lower efficacy [12]. Triptans and ergotamine are prescribed for moderate to severe migraine attacks, although ergotamine safety is uncertain [12]. Overuse of acute therapy can lead to medication overuse headaches (MOHs), especially in patients who suffer from more frequent and severe migraine attacks [13]. Patients who meet the criteria for prophylactic therapy can benefit greatly from the recent advances and innovations in the prevention of migraine attacks [13].

Calcitonin gene-related peptide (CGRP) is a neuropeptide found in the central and peripheral nervous system, which plays a significant role in the development of migraines through pain modulation in the trigeminovascular system [14]. Studies have shown that CGRP levels are remarkably elevated during migraine attacks, which is seen as a specific target to antagonize for the relief and prophylaxis of migraine [14,15,16]. For the purpose of improving the quality of life for patients affected by migraine, prophylactic medications are recommended to effectively decrease the frequency, duration, and intensity of migraine [16]. In comparison to traditional prophylactic medications such as beta-blockers, antiepileptics, calcium channel blockers, or 5-HT receptor antagonists, CGRP inhibitors offer better tolerability due to their selective mechanisms of action [17]. CGRP receptor antagonists (rimegepant and atogepant) and monoclonal antibodies against CGRP neuropeptide (eptinezumab, fremanezumab, and galcanezumab) or its receptor (erenumab) have fewer side effects [18]. The most frequently reported side effects are pain, redness, or swelling at the injection site, and constipation [19]. Nevertheless, it is essential to recognize the possible adverse effects of these newer medications as many of them are considered serious, including hypertension for patients treated with erenumab (Aimovig) [20,21].

As the demand for healthcare in society increases, community pharmacists play an essential role in improving patient adherence by providing medication-related information and counseling [22]. Effective counseling strategies include open-ended questions and active listening to ease the exchange of information between pharmacists and patients [22]. Community pharmacists frequently face limited time and patient interest challenges. Studies show many pharmacists spend only a few minutes counseling patients during the dispensing process, reducing the effectiveness of their interaction [23]. Community pharmacists are easily accessible. Therefore, they are the first choice for patients seeking relief from migraines [24]. Pharmacists can provide adequate patient education on migraine management, including identifying symptoms and triggers and suggesting lifestyle modifications [24]. They can recommend appropriate OTC medication, such as acetaminophen and NSAIDs, and guide prescription medication to minimize the risk of MOH, which can contribute to improving therapeutic outcomes in both acute and preventive treatment of migraine headaches [24]. In Saudi Arabia, community pharmacists’ roles have expanded beyond dispensing to include counseling, chronic disease support, and patient education [25]. However, challenges include workload pressures, access to professional development, and gaps in professional development [26]. Importantly, CGRP inhibitors are prescription-only medicines, and injectable formulations are not administered in community pharmacies but dispensed for use in clinical settings.

Moreover, community pharmacists can identify patients suffering from migraines and candidates for preventive treatment and address them to a specialist physician [24]. Therefore, CGRP inhibitors emerge as a novel and promising class of migraine therapies; it is critical to assess pharmacists’ knowledge, attitudes, and practices related to their use. Identifying existing gaps can guide the development of targeted educational interventions and continuing professional development programs. This study aims to evaluate the current knowledge, attitudes, and practices of pharmacists in Saudi Arabia toward CGRP inhibitors, with the ultimate goal of enhancing migraine management and optimizing patient outcomes. 

## 2. Methodology

### 2.1. Research Design, Participants, and Survey Instrument

We conducted a cross-sectional study to assess the knowledge, attitudes, and practices of pharmacists in Saudi Arabia regarding CGRP inhibitors. Data was collected via an online survey developed in English using Google Forms, which included 26 questions divided into four sections: demographics, knowledge, attitudes, and practices. The questionnaire was developed after a comprehensive review of the existing literature on pharmacists’ knowledge, attitudes, and practices regarding migraine management and CGRP inhibitors [24,27,28]. Content validity was established through expert review by two clinical pharmacy faculty members. The instrument was piloted with 15 pharmacists to assess clarity, relevance, and time to completion; minor adjustments were made accordingly. The final version consisted of four sections (demographics, knowledge, attitudes, and practices). A copy of the questionnaire is provided as Appendix A. The survey was conducted between October 2024 and January 2025 and distributed through direct visits to 135 local pharmacies in the western region (Makkah and Jeddah), and internal communication networks of regional and international pharmacy chains. The sample size was estimated using Raosoft^®^ software (http://www.raosoft.com/samplesize.html) with a 95% confidence level and 5% margin of error. The calculation was informed by the reported national number of 21,648 community pharmacists in Saudi Arabia in 2022 [26], which yielded a minimum required sample of 377 participants, as shown in Appendix A. Although the national figure was used for estimation, recruitment in practice included both community and hospital pharmacists working in the western region. Accordingly, the findings reflect the perspectives of pharmacists across these two practice settings within this regional context.

### 2.2. Assessing Pharmacist Perspectives on CGRP Inhibitors

Respondents’ demographic information was collected through nine questions, including gender, level of education, the source of their pharmacy degree, practice setting, region of practice within Saudi Arabia, and years of experience. Three questions related to migraines were included in the demographic section asking which headache condition respondents most frequently encounter in the practice. Respondents were asked to select different options from a multiple-choice list (i.e., migraine, sinus headache, chronic headache, tension headache, do not know). Another question was about attending a headache/migraine management course. The last question was about the number of patients suffering from migraines monthly in the pharmacy.

The survey included six questions to assess respondents’ knowledge of CGRP inhibitors, including familiarity with the class of medications, the primary indication for their use, the mechanism of action, the inclusion of CGRP inhibitors as a first-line agent in migraine prevention, awareness of relevant guidelines and recommendations, and potential adverse drug reactions (ADRs). Respondents could answer each question with “Yes”,’ “No”, or “Do not know”. To assess attitudes toward CGRP inhibitors, respondents answered six questions regarding their beliefs toward the medication, including comparison with typical migraine treatments, the benefits and potential ADRs associated with CGRP inhibitors, knowledge among healthcare providers about the class of medicines, patients’ convenience, the impact on the patient’s quality of life, and the need for further research in this area. Responses were recorded on a Likert scale with the following options: “Strongly agree”, “Agree”, “Neutral”, “Disagree”, and “Strongly disagree”.

Respondents’ practices regarding CGRP inhibitors were assessed through five questions, including patient education on the proper use of the injectable forms and possible ADRs, consideration of patient symptoms and previous medical history, collaboration with other healthcare providers, and the incorporation of additional clinical information (i.e., ADRs, drug–drug interactions, contraindications). Responses were recorded on a Likert scale, as previously mentioned in the attitudes section.

### 2.3. Ethical Approval

The Biomedical & Research Committee at Umm Al-Qura University approved the study before it was conducted under the approval number (HAPO-02-K-012-2024-10-2259). Participants were informed of the study’s purpose and procedure on the consent form, and their participation was voluntary. Responses were collected anonymously to ensure confidentiality.

### 2.4. Statistical Analysis

Statistical analyses were performed using IBM SPSS version 29.0.2. Descriptive statistics for categorical variables were presented as frequencies and percentages. For scoring, responses to positively phrased items were coded as “Yes” = 1 and “No” = 0. Negatively worded items were reverse coded so that higher scores consistently indicated higher knowledge, more positive attitudes, or better practices. This approach ensured consistency across domains. Recoding responses computed the attitude and practice scores: strongly disagree = 1, disagree = 2, neutral = 3, agree = 4, strongly agree = 5. The mean score for each participant in each domain was then computed. Internal consistency was evaluated using Cronbach’s Alpha for each domain, while construct validity was assessed through exploratory factor analysis. Relationships between domains were examined using Spearman’s rho correlation with two-tailed significance testing (*p* < 0.05 considered statistically significant). The internal consistency of the questionnaire was assessed using Cronbach’s Alpha, with values ≥0.70 considered acceptable, 0.80–0.89 considered good, and ≥0.90 considered excellent reliability. Regression analysis was performed to identify potential associations between demographic variables and the three domains under investigation.

## 3. Results

### 3.1. Demographic Characteristics

Of the 424 pharmacists approached, 419 agreed to participate (response rate = 98.8%). While this rate appears high, it should be interpreted with caution as recruitment was conducted through convenience sampling, which may introduce bias. The 419 respondents represent a fraction of the total community pharmacy workforce in Saudi Arabia, estimated at approximately 25,000 pharmacists [29]. Compared to national workforce demographics, our sample included a higher proportion of male pharmacists (75.9% vs. ~60% nationally) and a greater share from the western region. This reflects both sociocultural patterns and the sampling approach used.

Most participants graduated from foreign universities (62.1%), while 37.9% obtained their degrees from local Saudi universities. Most respondents work in a community pharmacy setting (90.5%), while 9.5% work in a hospital setting. Most of the pharmacists reported that sinus headaches were the most commonly seen condition in their practices at 65.2%. This was followed by tension headaches at 42.2%, migraines at 41.5%, and chronic headaches at 18.1%. More than half of the participants (52.7%) attended an educational course on managing headaches and migraines. When asked about the number of patients who visit the pharmacy suffering from migraine monthly, 43.2% of pharmacists reported having fewer than 10 patients, 31.0% reported 11–20 patients, 11.2% reported 21–30 patients and 14.6% reported more than 30 patients (Table 1).

### 3.2. Knowledge of CGRP Inhibitors

Over half of the participants were familiar with CGRP inhibitors (54.7%); 33.4% reported being unfamiliar, and 11.9% were unsure. Regarding the medication’s indications, 55.6% of the respondents were familiar with it, and 31.3% reported being unfamiliar. In response to a question about the mechanism of action of CGRP inhibitors in the management of migraines, 49.9% of the respondents were familiar, and 36.0% were unfamiliar. Around 44.6% of the participants reported being aware of the recent addition of CGRP inhibitors to first-line agents in migraine prophylaxis, and 37.0% were unaware. More than 44% of the participants were unfamiliar with the current guidelines (without specifying a particular guideline) for using CGRP inhibitors; only 37.2% were familiar. Almost 49% of the participants were unsure about the potential side effects of CGRP inhibitors; only 32.5% were familiar (Table 2).

### 3.3. Attitudes Toward CGRP-Based Migraine Management

When asked whether CGRP inhibitors are better for migraine management than typical medications, 48.9% of the respondents were neutral, and 32.2% agreed. About 56.1% were neutral regarding the benefits of CGRP inhibitors outweighing the potential side effects, and 27.9% agreed. The fact that there is a lack of knowledge about CGRP inhibitors among health providers was agreed upon by 38.4%, and 36% were neutral. In regard to the convenience of the injectable forms of CGRP inhibitors for the patients, 46.3% of the respondents were neutral, and 32% agreed. Concerning the improvement of quality of life in patients using CGRP inhibitors, 40.8% of the participants were neutral, and 40.3% expressed agreement. More research and clinical trials are needed to fully understand the benefits and possible long-term side effects of CGRP inhibitors, which were agreed upon by 36.8%, neutral by 39.4%, and strongly agreed upon by 17.4% (Table 3).

### 3.4. Practices Related to CGRP Inhibitor Use

In order to examine the practice patterns associated with CGRP inhibitors, pharmacists were asked about educating patients on the appropriate use of CGRP inhibitors; almost half of the pharmacists’ opinions were neutral (55.4%), and 28.4% agreed. Around 48.9% of the pharmacists exhibited a neutral response when asked about collaborating with other healthcare professionals for migraine management with CGRP inhibitors, and 32.7% agreed. When asked about acquiring the patient’s history and symptoms of their migraine type before dispensing their CGRP inhibitor medication, 39.9% of the participants exhibited a neutral response, and 37.9% agreed. Regarding seeking additional clinical information (ADRs, drug–drug interactions, and contraindications) before dispensing CGRP inhibitors, 42% had a neutral response, and 34.8% agreed. Pharmacists were asked whether they inform patients taking CGRP inhibitors about the possible side effects; 41.8% were neutral, and 38.9% agreed (Table 4).

### 3.5. Reliability, Validity, and Correlation Analysis of the Questionnaire

The psychometric properties of the questionnaire were evaluated to ensure reliability and validity. Table 5 presents the internal consistency results, showing that Cronbach’s Alpha values ranged from acceptable to good across the domains (Knowledge: α = 0.788, Attitude: α = 0.829, Practice: α = 0.846). This indicates that the instrument demonstrated satisfactory reliability for assessing pharmacists’ knowledge, attitudes, and practices toward CGRP inhibitors.

Construct validity was examined using exploratory factor analysis (EFA). As shown in Table 6, items clustered according to their respective domains, with strong factor loadings observed for Knowledge (items 10–15), Attitude (items 16–21), and Practice (items 22–26). These results confirm that the questionnaire adequately captured the underlying constructs and was appropriate for use in this study.

Furthermore, correlation analysis between the three constructs demonstrated statistically significant positive associations (Table 7). Knowledge scores were positively correlated with Attitude (ρ = 0.207, *p* < 0.001) and Practice (ρ = 0.249, *p* < 0.001). The strongest correlation was observed between Attitude and Practice (ρ = 0.529, *p* < 0.001). These results suggest that greater knowledge is associated with more positive attitudes and improved practices, and that attitudes strongly influence pharmacists’ practices regarding CGRP inhibitors.

### 3.6. Regression Analysis of Factors Influencing Knowledge, Attitude, and Practice

Multiple linear regression analyses were conducted to examine factors associated with pharmacists’ knowledge, attitudes, and practices toward CGRP inhibitors. The knowledge model (Table 8) was significant (R^2^ = 0.093, F = 5.236, *p* < 0.001), showing that attending a course on headache or migraine management was the only significant positive predictor (B = 0.178, *p* < 0.001). For attitudes (Table 9), the model was also significant (R^2^ = 0.048, F = 2.588, *p* = 0.009), with the female gender being positively associated with higher scores (B = 0.326, *p* = 0.002), while obtaining a pharmacy degree outside Saudi Arabia was negatively associated (B = −0.273, *p* = 0.021). The practice model (Table 10) demonstrated the strongest explanatory power (R^2^ = 0.109, F = 6.268, *p* < 0.001), with the female gender predicting higher practice scores (B = 0.393, *p* < 0.001), whereas pharmacists who graduated from institutions outside Saudi Arabia reported significantly lower practice scores (B = −0.413, *p* < 0.001).

## 4. Discussion

Migraine is one of the most prevalent neurological disorders with a wide variation in demographic and geographical aspects [30]. As reported in the literature in 2022, migraine prevalence in the United States was estimated to be around 22.7% [12] and 25–37.2% in Saudi Arabia [31,32]. When it comes to migraine treatment options, recent advances in the field have revealed that CGRP inhibitors offer a significant preventive advantage [30]. Thus, pharmacists play a crucial role in managing migraines through informed drug selection and effective patient education [30]. This study aimed to examine the knowledge, attitude, and practice of pharmacists regarding the use of CGRP inhibitors in the management and prophylaxis of migraine.

Participants reported frequent encounters with patients who self-describe their symptoms as ‘sinus headaches’ (65.2%). We emphasize that this reflects patient self-report rather than pharmacist diagnosis. In the context of CGRP inhibitors, such presentations highlight the need for pharmacists to recognize migraine-consistent features, provide appropriate counseling, and refer patients to prescribers when indicated, particularly given the moderate knowledge and limited guideline familiarity observed. A study conducted in Turkey evaluated the awareness of migraine among primary healthcare practitioners and reported that only 10.5% of the participants were able to accurately provide the complete diagnostic criteria for migraine without aura [33]. In Saudi Arabia, similar gaps have also been reported among physicians, who showed limited awareness of diagnostic criteria and optimal management strategies for migraine [34]. The majority of the participating healthcare practitioners failed to identify vital information like the lower limits of Chronic Migraine (CM) duration, the necessity of determining at least eight migraine attacks in a month, the importance of managing medication overuse, and the indication of topiramate as the most efficacious agent in CM [34]. Therefore, this study emphasizes the need for educational programs on migraines targeted specifically at pharmacists.

This study revealed that pharmacists showed moderate knowledge regarding preventive migraine treatments, including CGRP inhibitors, with overall scores of 54.7% for knowledge, 55.6% for attitudes, and 49.9% for practice. Training differences likely contribute to these results. For instance, in the U.S., roughly half of PharmD programs incorporate headache management content consistent with headache guideline recommendations [30,35]. Similar emphasis should be performed in local curricula, which may help the knowledge gaps observed. Community pharmacists serve as frontline counselors for migraine patients, particularly since many patients avoid physician visits and instead seek OTC guidance [36]. Despite their accessibility and potential to counsel on medication overuse and self-care, our findings suggest that knowledge of updated migraine therapies among community pharmacists remains limited, possibly due to insufficient continuing education and high workload. Hospital pharmacists contribute through oversight of medication use, especially in evaluating the appropriateness of OTC treatments and identifying cases warranting preventive therapy [30]. However, this study indicated wide variability in practice approaches and a neutral attitude toward interprofessional collaboration, suggesting structural barriers such as limited interdisciplinary communication or institutional support.

Although our findings show that 44.6% of participants were aware of the recent addition of CGRP inhibitors to first-line agents in migraine prevention, a surprisingly significant percentage (44.9%) had inadequate knowledge regarding the current guidelines for these medications. Clinical practice guidelines (CPGs) connect research with practice to provide evidence-based recommendations for all pharmacy professionals, with the purpose of achieving high-quality healthcare provision to patients [37]. Despite the emergence of guidelines, the recognition and implementation in actual practice are challenging [38]. A national study was conducted to assess the use of National Institute for Clinical Excellence (NICE) guidelines by physicians, hospital pharmacists, and other healthcare providers in the United Kingdom [39]. The results showed that the use of guidelines varied due to several barriers related to individuals’ attitudes and behaviors [39]. Another study in migraine management reported that only a few pharmacists make use of CPGs [40]. A systematic review of 25 studies identified a large number of different barriers. They classified the barriers into five categories: barriers related to social-political factors (e.g., difficulties with teamwork), barriers related to the health system (e.g., lack of time), barriers related to the CPGs (e.g., lack of clarity), barriers related to the healthcare practitioner (e.g., lack of knowledge about the CPG), and barriers related to patients (e.g., negative attitude towards implementation) [41]. Barriers must be identified in order to develop and apply strategies that encourage the healthcare system’s implementation of guidelines [42]. Access to CGRP inhibitors in Saudi Arabia is shaped by their high cost, restricted availability, and regulatory constraints. These therapies are typically administered through specialized headache clinics supported by the Ministry of Health, limiting distribution primarily to tertiary settings [43]. Due to their expense often ranking among the high-cost migraine treatments, many patients may face financial barriers, especially if insurance coverage is inadequate [44]. Thus, these economic factors likely contribute to pharmacists’ limited exposure and familiarity with CGRP inhibitors in community settings, underscoring the need for broader access strategies and updated regulatory frameworks.

According to our findings, 32.2% of pharmacists agreed that CGRP inhibitors are more effective than typical medications for migraine management. Studies have shown that CGRP inhibitors are more effective than traditional medications and are well-tolerated, with few side effects [45,46]. Furthermore, 27.9% of participants believed that the benefits of CGRP inhibitors outweigh the potential side effects, while 56.1% remained neutral. When comparing CGRP inhibitors to older preventive options (e.g., beta-blockers, anticonvulsants, antidepressants), CGRP inhibitors demonstrate a favorable benefit-to-risk ratio of approximately 10–25 (high benefit, low harm), whereas older preventive treatments show a ratio of around 0.5–4 (moderate benefit, higher harm). These findings establish CGRP inhibitors as a preferred option for patients who experience tolerability and efficacy challenges with older preventive medications [47]. As well, 38.4% of participants acknowledged a significant knowledge gap about CGRP inhibitors among healthcare providers. This finding aligns with a study that assessed primary healthcare practitioners’ knowledge, attitudes, and practices regarding the management of migraine in Saudi Arabia. Approximately 70.8% of the physicians participating in the study were unaware of the newly approved preventive treatment of CGRP inhibitors [34].

Although several previous studies have highlighted the important role of pharmacists in patient education and medication adherence, especially in migraine management, this study found that 55.4% of responders exhibit a neutral response, while only 28.4% agreed when asked whether they educate patients on the proper use of CGRP inhibitors [24,30,48]. Likewise, most respondents displayed neutrality in obtaining patients’ histories and symptoms, seeking information on adverse drug reactions and interactions, and communicating potential side effects to patients. This finding suggests a potential knowledge gap or a lack of structured training on newer agents.

In addition, it has been reported that pharmacists play a crucial role in contributing to achieving optimal therapeutic outcomes and improving patients’ quality of life [49]. However, our findings indicate that 48.9% of pharmacists reported neutral responses toward collaborating with other healthcare providers, whereas 32.7% agreed. This missed opportunity for effective management could be due to several factors, such as inadequate organizational support (including time limitations and a lack of professional training), poor communication, and interpersonal barriers [50]. On the other hand, interdisciplinary collaboration is essential for optimal migraine care, particularly with new preventive options like CGRP inhibitors. However, our findings of pharmacists’ neutral attitudes toward collaboration likely reflect systemic challenges. For example, technological barriers including limited access to shared electronic health records can delay information exchange and workflow efficiency between pharmacists and physicians [51,52].

This study has several limitations. An imbalance between males and females was observed, as male pharmacists represented the majority of the total participants. The imbalance is demonstrated in the overall knowledge, attitude, and practice. This observation may be a demonstration of the sociocultural factors present in Saudi Arabia. Regardless of the recent gradual increase in female pharmacists, the field is still considered to be male-dominated [29,53,54,55,56]. One of the attitude questions combined two elements (perceived benefits and long-term safety of CGRP inhibitors). This may have introduced ambiguity in interpretation, as agreement could reflect either or both aspects. Furthermore, the study’s findings may be limited by the fact that the majority of participants are from the western region of Saudi Arabia, with a more significant portion of the participants from the community pharmacy setting. In addition, the use of convenience sampling may have introduced selection bias, and thus the findings should be interpreted with caution when generalizing to the wider population of pharmacists in Saudi Arabia.

## 5. Conclusions

This study assessed pharmacists’ knowledge, attitudes, and practices regarding migraine management, particularly with the emergence of CGRP inhibitors as preventive treatments. While pharmacists showed moderate knowledge of these medications, notable gaps exist in guideline awareness, patient education, and interdisciplinary collaboration. The frequent self-reporting of migraine as ‘sinus headache’ highlights the importance of pharmacists in recognizing potential migraine features and referring patients for proper medical diagnosis. Targeted educational programs and training are essential to bridge these gaps and ensure pharmacists are equipped to support optimal migraine care and improve patient outcomes.

## Figures and Tables

**Table 1 healthcare-13-02231-t001:** Demographics of Participating Pharmacists.

Variable	Frequency (N = 419)	Percentage
Gender		
Male	318	75.9
Female	101	24.1
Level of Education		
BSc	238	56.8
PharmD	153	36.5
Master’s	18	4.3
PhD	10	2.4
Source of Pharmacy Degree		
Foreign University	260	62.1
Saudi University	159	37.9
Practice Setting		
Community Pharmacy	379	90.5
Hospital Pharmacy	40	9.5
Region of practice in Saudi Arabia		
Western	202	48.2
Eastern	68	16.2
Central	80	19.1
Southern	42	10.0
Northern	27	6.4
Years of Experience		
Less than 1 year	58	13.8
1–5 years	131	31.3
6–10 years	94	22.4
More than 10 years	136	32.5
Which headache condition do you most frequently encounter in your practice?		
Migraine	174	41.5
Sinus headache	273	65.2
Chronic headache	76	18.1
Tension headache	177	42.2
Do not know	16	3.8
Have you attended a course about headache/migraine management	221	52.7
Number of patients visiting your pharmacy suffering from migraines monthly		
Less than 10 patients	181	43.2
11–20 patients	130	31.0
21–30 patients	47	11.2
More than 30 patients	61	14.6

**Table 2 healthcare-13-02231-t002:** Pharmacists’ Knowledge of CGRP Inhibitors.

Question	No		Yes		Do Not Know	
	N	%	N	%	N	%
Are you familiar with Calcitonin Gene-Related Peptide (CGRP) inhibitors (e.g., erenumab (Aimovig)) as a class of medication?	140	33.4	229	54.7	50	11.9
Do you know what are the main indications for CGRP-inhibitors?	131	31.3	233	55.6	55	13.1
Do you know how CGRP inhibitors work to treat migraines?	151	36.0	209	49.9	59	14.1
Are you aware that CGRP inhibitors have recently been added to the first-line agents in migraine prevention?	155	37.0	187	44.6	77	18.4
Are you familiar with the current guidelines or clinical recommendations in using CGRP inhibitor?	188	44.9	156	37.2	75	17.9
Is it true that CGRP inhibitors can cause an increase in blood pressure?	79	18.9	136	32.5	204	48.7

**Table 3 healthcare-13-02231-t003:** Pharmacists’ Attitudes Toward CGRP Inhibitors.

Question	Strongly Disagree (1)		Disagree (2)		Neutral (3)		Agree (4)		Strongly Agree (5)		Mean	Median
	N	%	N	%	N	%	N	%	N	%		
I believe CGRP inhibitors are a better choice for migraine management than typical medications.	29	6.9	17	4.1	205	48.9	135	32.2	33	7.9	3.3	3
I believe the benefits of CGRP inhibitors in migraine management outweigh the potential side effects.	23	5.5	22	5.3	235	56.1	117	27.9	22	5.3	3.2	3
I believe there is a lack of knowledge about CGRP inhibitors among health providers.	22	5.3	18	4.3	151	36	161	38.4	67	16	3.6	4
I believe the injectable form of CGRP inhibitors is convenient for patients.	25	6	48	11.5	194	46.3	134	32	18	4.3	3.2	3
I believe that CGRP inhibitors improve the quality of life for migraine patients.	16	3.8	11	2.6	171	40.8	169	40.3	52	12.4	3.6	4
I believe that more research and clinical trials are needed to fully understand the benefits and possible long-term side effects of CGRP inhibitors.	19	4.5	8	1.9	165	39.4	154	36.8	73	17.4	3.6	4

**Table 4 healthcare-13-02231-t004:** Pharmacists’ Practices Regarding CGRP Inhibitor Use.

Question	Strongly Disagree (1)		Disagree (2)		Neutral (3)		Agree (4)		Strongly Agree (5)		Mean	Median
	N	%	N	%	N	%	N	%	N	%		
I educate patients on the appropriate use of CGRP inhibitors.	26	6.2	22	5.3	232	55.4	119	28.4	20	4.8	3.2	3
I collaborate with other healthcare professionals (physicians/neurologists) for migraine management with CGRP inhibitor medications.	22	5.3	33	7.9	205	48.9	137	32.7	22	5.3	3.3	3
I ask about the patient’s history and symptoms of their migraine type before deciding to dispense their CGRP inhibitor medication.	15	3.6	22	5.3	167	39.9	159	37.9	56	13.4	3.5	4
I seek additional clinical information (ADRs, DDIs, C/I) before dispensing CGRP inhibitors	17	4.1	25	6	176	42	146	34.8	55	13.1	3.5	3
I inform patients taking CGRP inhibitors about the possible side effects such as allergic reactions, constipation, and high blood pressure.	20	4.8	14	3.3	175	41.8	163	38.9	47	11.2	3.5	4

**Table 5 healthcare-13-02231-t005:** Internal Consistency Analysis.

Item	Cronbach’s Alpha	Interpretation
Knowledge	0.788	Acceptable
Attitude	0.829	Good
Perception	0.846	Good

**Table 6 healthcare-13-02231-t006:** Construct Validity Analysis by Exploratory Factor Analysis.

Question Number	Question	Factor Loading on Components		
		Attitude	Knowledge	Practice
10	Are you familiar with Calcitonin Gene-Related Peptide (CGRP) inhibitors (e.g., erenumab (Aimovig)) as a class of medication?	0.071	0.721	−0.052
11	Do you know what are the main indications for CGRP-Inhibitors?	0.055	0.804	0.023
12	Do you know how CGRP inhibitors work to treat migraines?	−0.022	0.798	0.088
13	Are you aware that CGRP inhibitors have recently been added to the first-line agents in migraine prevention?	−0.027	0.775	0.011
14	Are you familiar with the current guidelines or clinical recommendations in using CGRP inhibitor?	−0.104	0.784	0.068
15	Is it true that CGRP inhibitors can cause an increase in blood pressure?	0.264	0.289	−0.253
16	I believe CGRP inhibitors are a better choice for migraine management than typical medications.	0.737	0.016	0.216
17	I believe the benefits of CGRP inhibitors in migraine management outweigh the potential side effects.	0.721	0.046	0.221
18	I believe there is a lack of knowledge about CGRP inhibitors among health providers.	0.654	−0.075	0.206
19	I believe the injectable form of CGRP inhibitors is convenient for patients.	0.560	0.077	0.150
20	I believe that CGRP inhibitors improve the quality of life for migraine patients.	0.808	−0.019	0.262
21	I believe that more research and clinical trials are needed to fully understand the benefits and possible long-term side effects of CGRP inhibitors.	0.679	−0.059	0.305
22	I educate patients on the appropriate use of CGRP inhibitors.	0.333	0.009	0.611
23	I collaborate with other healthcare professionals (physicians/neurologists) for migraine management with CGRP inhibitor medications.	0.259	0.125	0.736
24	I ask about the patient’s history and symptoms of their migraine type before deciding to dispense their CGRP inhibitor medication.	0.218	0.033	0.821
25	I seek additional clinical information (ADRs, DDIs, C/I) before dispensing CGRP inhibitors	0.268	−0.048	0.730
26	I inform patients taking CGRP inhibitors about the possible side effects such as allergic reactions, constipation, and high blood pressure.	0.276	0.018	0.752

**Table 7 healthcare-13-02231-t007:** Correlation Analysis between the 3 constructs.

		Knowledge Mean Score	Attitude Mean Score	Practice Mean Score
Knowledge Mean Score	Spearman’s rho	-	0.207	0.249
	*p*-value	-	<0.001	<0.001
Attitude Mean Score	Spearman’s rho	-	-	0.529
	*p*-value	-	-	<0.001
Practice Mean Score	Spearman’s rho	-	-	-
	*p*-value	-	-	-

**Table 8 healthcare-13-02231-t008:** Correlation Multiple Linear Regression Analysis of Predictors of Pharmacists’ Knowledge Toward CGRP Inhibitors (R^2^ = 0.093, F = 5.236, *p* < 0.001).

Variable	B	t	*p*-Value	95% Confidence Interval	
				Lower	Upper
Gender	−0.027	−0.493	0.623	−0.137	0.082
Level of Education	−0.035	−1.317	0.188	−0.087	0.017
Source of Pharmacy Degree	0.008	0.138	0.891	−0.112	0.129
Practice Setting	−0.028	−0.422	0.673	−0.157	0.101
Region of Practice in Saudi Arabia	0.014	0.995	0.320	−0.013	0.040
Years of Experience	0.022	0.939	0.348	−0.024	0.069
Have you attended a course about headache/migraine management	0.178	5.151	<0.001	0.110	0.247
Number of patients visiting your pharmacy suffering from migraines monthly	0.002	0.113	0.910	−0.030	0.034

**Table 9 healthcare-13-02231-t009:** Correlation Multiple Linear Regression Analysis of Predictors of Pharmacists’ Attitudes Toward CGRP Inhibitors (R^2^ = 0.048, F = 2.588, *p* = 0.009).

Variable	B	t	*p*-Value	95% Confidence Interval	
				Lower	Upper
Gender	0.326	3.057	0.002	0.116	0.536
Level of Education	−0.045	−0.875	0.382	−0.145	0.055
Source of Pharmacy Degree	−0.273	−2.321	0.021	−0.504	−0.042
Practice Setting	0.010	0.078	0.938	−0.238	0.258
Region of Practice in Saudi Arabia	−0.051	−1.954	0.051	−0.103	0.000
Years of Experience	0.039	0.852	0.394	−0.051	0.128
Have you attended a course about headache/migraine management	−0.014	−0.205	0.838	−0.144	0.117
Number of patients visiting your pharmacy suffering from migraines monthly	−0.003	−0.088	0.930	−0.064	0.059

**Table 10 healthcare-13-02231-t010:** Multiple Linear Regression Analysis of Predictors of Pharmacists’ Practices Toward CGRP Inhibitors (R^2^ = 0.109, F = 6.268, *p* < 0.001).

Variable	B	t	*p*-Value	95% Confidence Interval	
				Lower	Upper
Gender	0.393	3.623	<0.001	0.180	0.606
Level of Education	−0.033	−0.646	0.519	−0.135	0.068
Source of Pharmacy Degree	−0.413	−3.455	<0.001	−0.648	−0.178
Practice Setting	−0.176	−1.376	0.170	−0.428	0.076
Region of Practice in Saudi Arabia	−0.049	−1.841	0.066	−0.101	0.003
Years of Experience	0.071	1.540	0.124	−0.020	0.162
Have you attended a course about headache/migraine management	0.085	1.254	0.210	−0.048	0.218
Number of patients visiting your pharmacy suffering from migraines monthly	0.039	1.218	0.224	−0.024	0.101

## Data Availability

The original contributions presented in this study are included in the article. Further inquiries can be directed to the corresponding author.

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
