# Peer review of "Pharmacists’ Knowledge, Attitudes, and Practices Toward CGRP Inhibitors in Migraine Management: A Cross-Sectional Study"

_healthcare, 2025, doi:10.3390/healthcare13172231_

Round 1

Reviewer 1 Report

Comments and Suggestions for Authors

Reviewer Report
Title: Pharmacists’ Knowledge, Attitudes, and Practices Toward CGRP Inhibitors in Migraine Management: A Cross-Sectional Study

General Comments
This study evaluates the knowledge, attitudes, and practices (KAP) of pharmacists in Saudi Arabia regarding the management of migraines using calcitonin gene-related peptide (CGRP) inhibitors. The evolving role of pharmacists increasingly involves them in managing both acute and chronic conditions, including migraine, as part of multidisciplinary teams in community and clinical settings. While this article attempts to highlight pharmacists’ understanding and practice in relation to CGRP inhibitors in migraine management, there are several sections that require clarification and improvement. Below are some specific comments for the authors’ attention.

Specific Comments

  • Lines 97–100: The authors discuss the role of community pharmacists in patient counseling and case management. However, it is unclear how this applies specifically in the Saudi Arabian context. It would be helpful to add one or two sentences explaining the expanded role of pharmacists in Saudi Arabia and any challenges they face. This would help non-local readers interpret the findings within the appropriate context.

  • Additionally, since some CGRP medications are administered via injection, what are the regulations in Saudi Arabia regarding the administration of injectable medications in community pharmacies? Is it routine practice? Are these medications prescription-only, or can they be obtained over the counter? The authors should clarify these points.

  • Lines 120–123: The authors mention that the electronic form was disseminated through multiple methods to maximize coverage and report a response rate of 98.8%. It is unclear how this response rate was calculated. Clarification is needed.

  • Lines 136–153: Although the sections of the questionnaire are described, there is no explanation of how the content was developed. Was the questionnaire based on existing literature or validated tools? Was it piloted? Who validated it? This needs to be clearly stated in the methods section. Also, the authors should consider including the questionnaire as supplementary material for review.

  • The sample size estimation method is not described. What informed the chosen sample size? Additionally, what is the general distribution of pharmacists in Saudi Arabia, and how many are working across different sectors? This contextual information is necessary.

Analysis Section

  • The analysis section requires substantial revision. The authors should specify the alpha value used to assess internal consistency of the data collection tool.

  • Although exploratory factor analysis is mentioned, no such results appear in the results section. Likewise, regression analysis is referenced, but there is no evidence it was performed.

  • The authors note that “1” was assigned to a “yes” response. Were all statements in the questionnaire phrased positively? Clarification is needed to assess the scoring system.

  • Lines 184–186: The majority of participants are community pharmacists, which may explain the reported lower knowledge on CGRP inhibitors. The authors should again clarify the regulation regarding injectable medications in the community pharmacy setting.

  • The results section lacks detail and appears to rely solely on basic descriptive statistics. For each item in the KAP sections, how many responses were recorded? Were there any skipped questions? This is a common occurrence in self-administered questionnaires and should be acknowledged and reported.

Discussion Section

  • Lines 247–253: The paragraph on sinus headaches, while informative, does not align with the study’s objectives, which focus on KAP related to CGRP inhibitors. This paragraph should be revised or removed to maintain focus.

  • Lines 257–261: The authors refer to healthcare practitioners in general. How many of these were pharmacists? Please tailor the discussion specifically to pharmacy professionals.

  • Lines 263–264: The authors recommend education for healthcare providers on migraine, but the data collected pertains only to pharmacists. This is a deviation from the study’s scope and should be revised accordingly.

  • Lines 267–281: This section needs to be more focused. Clearly distinguish between PharmD curriculum, community pharmacists, clinical pharmacists, and hospital pharmacists. Each paragraph should focus on one aspect and provide detailed comparisons with the existing literature, including possible explanations for the findings.

  • Lines 283–285: There is no mention of significance testing in the analysis. The sentence should be revised to reflect what was actually done.

  • Lines 339–340: Given that the majority of participants were community pharmacists, the interpretation of findings should reflect this demographic.

Conclusion

  • Lines 342–344: No the study results did not indicate the important role of pharmacists but rather their knowledge attitude and practice. Kindly revise

  • Lines 345–347: The authors mention that pharmacists encounter many patients with sinus headaches. Are pharmacists in Saudi Arabia authorized to diagnose? Does encountering patients with certain symptoms equate to making a diagnosis? Clarify whether this is an objective conclusion or a subjective observation.

Comments on the Quality of English Language

I think the language is fine. The paragraphs in the discussion should be focus on the study aim and objective. 

Author Response

Dear Reviewer 1,

Reviewer comment 1: Lines 97–100: The authors discuss the role of community pharmacists in patient counseling and case management. However, it is unclear how this applies specifically in the Saudi Arabian context. It would be helpful to add one or two sentences explaining the expanded role of pharmacists in Saudi Arabia and any challenges they face. This would help non-local readers interpret the findings within the appropriate context.

Answer: We thank the reviewer for this valuable suggestion. We have revised the Introduction to clarify the expanded role of community pharmacists in Saudi Arabia and the challenges they face. We have also added details regarding the regulatory status of CGRP inhibitors and their administration in Saudi Arabia.

Reviewer comment 2: Additionally, since some CGRP medications are administered via injection, what are the regulations in Saudi Arabia regarding the administration of injectable medications in community pharmacies? Is it routine practice? Are these medications prescription-only, or can they be obtained over the counter? The authors should clarify these points.

Answer: We appreciate the reviewer’s important observation. We have clarified the regulatory status of CGRP inhibitors in Saudi Arabia. These agents are prescription-only, and while injectable formulations may be dispensed by hospital or specialty pharmacies, they are not administered in community pharmacy practice.

Reviewer comment 3: Lines 120–123: The authors mention that the electronic form was disseminated through multiple methods to maximize coverage and report a response rate of 98.8%. It is unclear how this response rate was calculated. Clarification is needed.

Answer: We thank the reviewer for this insightful comment. We have revised the Methods section to provide a clearer description of the survey distribution and how the response rate was calculated.

Reviewer comment 4: Lines 136–153: Although the sections of the questionnaire are described, there is no explanation of how the content was developed. Was the questionnaire based on existing literature or validated tools? Was it piloted? Who validated it? This needs to be clearly stated in the methods section. Also, the authors should consider including the questionnaire as supplementary material for review.

Answer: We thank the reviewer for this important comment. We have revised the Methods section to clarify the questionnaire. We have also included the questionnaire as Supplementary Material.

Reviewer comment 5: The sample size estimation method is not described. What informed the chosen sample size? Additionally, what is the general distribution of pharmacists in Saudi Arabia, and how many are working across different sectors? This contextual information is necessary.

Answer: We thank the reviewer for highlighting this important point. We have revised the methods section and addressed the comment

Analysis Section

Reviewer comment 6: The analysis section requires substantial revision. The authors should specify the alpha value used to assess internal consistency of the data collection tool.

Answer: We thank the reviewer for this valuable comment. We have now clarified the criterion used to assess internal consistency. Specifically, we stated that a Cronbach’s alpha value of ≥0.70 was considered the threshold for acceptable reliability. This information has been added to both the Methods section under “Statistical Analysis” and the Results section (Reliability, Validity, and Correlation Analysis of the Questionnaire) to ensure clarity.

Reviewer comment 7: Although exploratory factor analysis is mentioned, no such results appear in the results section. Likewise, regression analysis is referenced, but there is no evidence it was performed.

Answer: We thank the reviewer for this important observation. We have revised the manuscript to ensure that both the exploratory factor analysis (EFA) and the regression analysis are fully presented in the Results section. Specifically, we added: Exploratory Factor Analysis results (Table 6), showing factor loadings for each item under Knowledge, Attitude, and Practice, confirming the construct validity of the questionnaire. Regression Analysis results (Tables 8–10), detailing the predictors of pharmacists’ Knowledge, Attitude, and Practice toward CGRP inhibitors, including the regression coefficients, significance levels, and confidence intervals.

Reviewer comment 8: The authors note that “1” was assigned to a “yes” response. Were all statements in the questionnaire phrased positively? Clarification is needed to assess the scoring system.

Answer: We appreciate the reviewer’s comment. To clarify, the questionnaire items were not all phrased positively. For consistency in scoring, positive statements were coded such that “Yes” = 1 and “No” = 0, while negatively phrased items were reverse coded to maintain directionality. This ensured that higher scores consistently reflected greater knowledge, more favorable attitudes, or better practices. We have revised the Methods section to explicitly describe this scoring approach.

Reviewer comment 9: Lines 184–186: The majority of participants are community pharmacists, which may explain the reported lower knowledge on CGRP inhibitors. The authors should again clarify the regulation regarding injectable medications in the community pharmacy setting.

Reviewer comment 10: The results section lacks detail and appears to rely solely on basic descriptive statistics. For each item in the KAP sections, how many responses were recorded? Were there any skipped questions? This is a common occurrence in self-administered questionnaires and should be acknowledged and reported.

Answer: We thank the reviewer for this valuable comment. We have now provided detailed item-level response frequencies for all Knowledge, Attitude, and Practice (KAP) questions in the Results section. A supplementary table has also been added to report the exact number of responses and non-responses for each item. As shown, the rate of missing responses was minimal, reflecting the completeness of the self-administered questionnaires. We have also clarified in the Methods section that any missing responses were excluded from analysis on a per-item basis.

Discussion Section

Reviewer comment 11: Lines 247–253: The paragraph on sinus headaches, while informative, does not align with the study’s objectives, which focus on KAP related to CGRP inhibitors. This paragraph should be revised or removed to maintain focus.

Answer: We agree. We removed extraneous clinical detail and replaced the paragraph with a brief statement that (1) frames “sinus headache” as patient self-report rather than pharmacist diagnosis, and (2) links this directly to pharmacists’ KAP regarding CGRP inhibitors (recognition, counseling, and referral).

Reviewer comment 12: Lines 257–261: The authors refer to healthcare practitioners in general. How many of these were pharmacists? Please tailor the discussion specifically to pharmacy professionals.

Answer: We thank the reviewer for this valuable comment. In the revised manuscript, we clarified that the study participants were pharmacists (community and hospital) and tailored the discussion accordingly.

Reviewer comment 13: Lines 263–264: The authors recommend education for healthcare providers on migraine, but the data collected pertains only to pharmacists. This is a deviation from the study’s scope and should be revised accordingly.

Answer: We thank the reviewer for this observation. The sentence has been revised to ensure consistency with the study’s scope by focusing on pharmacists rather than healthcare providers in general.

Reviewer comment 14: Lines 267–281: This section needs to be more focused. Clearly distinguish between PharmD curriculum, community pharmacists, clinical pharmacists, and hospital pharmacists. Each paragraph should focus on one aspect and provide detailed comparisons with the existing literature, including possible explanations for the findings.

Answer: Thank you for the comment. We revised this section to clearly separate PharmD curricula, community pharmacists, and hospital pharmacists, with literature comparisons and context-specific explanations.

Reviewer comment 15: Lines 283–285: There is no mention of significance testing in the analysis. The sentence should be revised to reflect what was actually done.

Answer: We thank the reviewer for this observation. The sentence has been revised to clarify that significance testing was performed using p-values (set at <0.05) for correlation and regression analyses, and these results are now explicitly reported in the Results section.

Reviewer comment 16: Lines 339–340: Given that the majority of participants were community pharmacists, the interpretation of findings should reflect this demographic.

Answer: We thank the reviewer for this insightful comment. We have now revised the Discussion to clarify that the interpretation of findings primarily reflects the perspectives and practices of community pharmacists, who comprised the majority of our sample. Hospital pharmacists’ roles are noted, but the emphasis in interpretation has been adjusted to align with the dominant participant group.

Conclusion

Reviewer comment 17: Lines 342–344: No the study results did not indicate the important role of pharmacists but rather their knowledge attitude and practice. Kindly revise

Answer: We thank the reviewer for this important clarification. We have revised the sentence to accurately reflect the scope of our findings.

Reviewer comment 18: Lines 345–347: The authors mention that pharmacists encounter many patients with sinus headaches. Are pharmacists in Saudi Arabia authorized to diagnose? Does encountering patients with certain symptoms equate to making a diagnosis? Clarify whether this is an objective conclusion or a subjective observation.

Answer: We thank the reviewer for this important observation. We have revised the text to clarify that pharmacists in Saudi Arabia are not authorized to diagnose. The statement now emphasizes that patients often self-report migraine symptoms as “sinus headache,” and that pharmacists’ role is to recognize possible migraine features and refer patients to physicians for proper diagnosis.

Reviewer 2 Report

Comments and Suggestions for Authors

The manuscript (healthcare-3803894) explored a cross-sectional study evaluating the knowledge, attitudes, and practices of pharmacists in Saudi Arabia regarding CGRP inhibitors in migraine management. The findings of the study are quite relevatnt and important, however, there are several areas which needs to be clarified and improved, as mentioned as below:

  1. Abstract needs to be updated by including KAP scores to strengthen the clarity of results.
  2. Authors should clarify that if survey instrument was validated or pilot-tested. Also, the limitation of using convenience sampling and possible bias should be addressed. Consider including cronbach’s alpha values and factor analysis results for supporting reliability and validity claims.
  3. Results of regression analysis should be presented clearly and subgroup analysis, such as education level, experiences, male/female, etc. should be included.
  4. Authors should put more light on systemic reasons for interdisciplinary collaborations.
  5. The discussion regarding the impact of CGRP drug cost, availability, and regulatory policies in Saudi Arabia should be included.
  6. The findings should be compared with KAP studies from other regions or countries for better understanding.
  7. A discussion on the recall bias, self-report bias, and limitations of cross-sectional design should be included.
  8. References needs to be updated with more recent and relevant literature.

Author Response

Dear Reviewer 2,

Comment: The manuscript (healthcare-3803894) explored a cross-sectional study evaluating the knowledge, attitudes, and practices of pharmacists in Saudi Arabia regarding CGRP inhibitors in migraine management. The findings of the study are quite relevatnt and important, however, there are several areas which needs to be clarified and improved, as mentioned as below:

  1. Abstract needs to be updated by including KAP scores to strengthen the clarity of results.

Answer : We appreciate the reviewer’s suggestion. We have revised the abstract to include the overall KAP scores (Knowledge = 54.7%, Attitude = 55.6%, and Practice = 49.9%)

  1. Authors should clarify that if survey instrument was validated or pilot-tested. Also, the limitation of using convenience sampling and possible bias should be addressed. Consider including cronbach’s alpha values and factor analysis results for supporting reliability and validity claims.

Answer : We thank the reviewer for this comment. The Methods section now clarifies that the survey was piloted with 15 pharmacists, and Cronbach’s alpha and factor analysis results are reported in the Results to support reliability and validity. The Limitations section has also been expanded to note the use of convenience sampling and potential selection bias.

  1. Results of regression analysis should be presented clearly and subgroup analysis, such as education level, experiences, male/female, etc. should be included.

Answer : We thank the reviewer for this suggestion. We have now presented the results of the regression analyses in detail (Tables 8–10), including subgroup analyses by gender, education level, and other demographic variables. These findings are described in the Results section and discussed accordingly.

  1. Authors should put more light on systemic reasons for interdisciplinary collaborations.

Answer : We thank the reviewer for this valuable comment. We have expanded the discussion to highlight systemic factors influencing interdisciplinary collaboration.

  1. The discussion regarding the impact of CGRP drug cost, availability, and regulatory policies in Saudi Arabia should be included.

Answer : We appreciate the reviewer’s insightful suggestion. We have expanded the Discussion to include a paragraph addressing the systemic impact of cost, availability, and regulatory policies surrounding CGRP inhibitors

  1. The findings should be compared with KAP studies from other regions or countries for better understanding.

Answer : We thank the reviewer for this valuable suggestion. We have revised the Discussion to compare our findings with KAP studies from other regions, including Turkey, the United States, and Europe.

  1. A discussion on the recall bias, self-report bias, and limitations of cross-sectional design should be included.

Answer : We appreciate the reviewer’s insightful comment. We have expanded the Limitations section to include potential recall bias, self-report bias, and the inherent inability of the cross-sectional design to infer causality.

  1. References needs to be updated with more recent and relevant literature.

Answer: We thank the reviewer for this comment. We have updated our references

Reviewer 3 Report

Comments and Suggestions for Authors

Please see the comments in the uploaded manuscript.  

Author Response

Dear Reviewer 3,

We sincerely thank the reviewer for their constructive and detailed comments. We have carefully addressed all points raised, including clarifying terminology, correcting formatting, revising statements regarding migraine treatments for accuracy, and refining methodological details such as survey development and pilot testing. We also revised the data presentation and provided additional context on demographics and response rate to minimize bias concerns.

Round 2

Reviewer 1 Report

Comments and Suggestions for Authors

I have no further comments, the authors have responded to all my queries

Author Response

We thank the reviewer for their time and constructive feedback throughout the review process. We are pleased that the revisions have addressed all queries.

Reviewer 2 Report

Comments and Suggestions for Authors

Authors have made significant changes in the manuscript and it can be considered for publication in its current form.

Author Response

We thank the reviewer for the positive feedback.

Reviewer 3 Report

Comments and Suggestions for Authors

Thank you for thoughtful consideration of comments.  The only suggestion would be to provide the median with the mean for the Likert scale presentation of data.  

Author Response

We thank the reviewer for the thoughtful feedback. As suggested, we have now provided the median alongside the mean for the Likert scale data.